# Advances in Understanding of the Copper Homeostasis in *Pseudomonas aeruginosa*

**DOI:** 10.3390/ijms22042050

**Published:** 2021-02-19

**Authors:** Lukas Hofmann, Melanie Hirsch, Sharon Ruthstein

**Affiliations:** Institute of Nanotechnology and Advanced Materials & Department of Chemistry, Faculty of Exact Sciences, Bar-Ilan University, Ramat-Gan 5290002, Israel; hofmmal@biu.ac.il (L.H.); melanie.13995@gmail.com (M.H.)

**Keywords:** *Pseudomonas aeruginosa*, copper homeostasis, multidrug resistance, pathogen, antibiotic resistance, transcription regulation, transcription factors, cueR, copRS, cusCBA

## Abstract

Thirty-five thousand people die as a result of more than 2.8 million antibiotic-resistant infections in the United States of America per year. *Pseudomonas aeruginosa* (*P. aeruginosa*) is classified a serious threat, the second-highest threat category of the U.S. Department of Health and Human Services. Among others, the World Health Organization (WHO) encourages the discovery and development of novel antibiotic classes with new targets and mechanisms of action without cross-resistance to existing classes. To find potential new target sites in pathogenic bacteria, such as *P. aeruginosa*, it is inevitable to fully understand the molecular mechanism of homeostasis, metabolism, regulation, growth, and resistances thereof. *P. aeruginosa* maintains a sophisticated copper defense cascade comprising three stages, resembling those of public safety organizations. These stages include copper scavenging, first responder, and second responder. Similar mechanisms are found in numerous pathogens. Here we compare the copper-dependent transcription regulators cueR and copRS of *Escherichia coli* (*E. coli*) and *P. aeruginosa*. Further, phylogenetic analysis and structural modelling of mexPQ-opmE reveal that this efflux pump is unlikely to be involved in the copper export of *P. aeruginosa*. Altogether, we present current understandings of the copper homeostasis in *P. aeruginosa* and potential new target sites for antimicrobial agents or a combinatorial drug regimen in the fight against multidrug resistant pathogens.

## 1. Introduction

### P. aeruginosa Origin, Occurrence, Risks, and Pathogenesis

*Pseudomonas aeruginosa* is a term created by the German botanist Walter Migula in 1894. *Pseudomonas* is a combination of the Greek words *pseudo* and *monas*, meaning “false” and “single unit,” respectively. This name was mistakenly derived from the resemblance to other bacteria discovered at this time. *Aeruginosa* from Latin means “copper rust or green,” given because of the greenish-blue color of the colonies [1]. *P. aeruginosa* colonizes plants, including vegetables or fruits, fungi, water, and soil. These strains are identical to the ones found in hospitals and patients [2,3]. Moreover, it has been shown that *P. aeruginosa* strains derived from hospitals are susceptible to competition by environmental pseudomonads [4]. This competition between hospital and environmental strains may indicate different defense mechanisms among pseudomonad strains. This difference is also observed in the relative sequence abundance of samples taken from different habitats that range from 0.1% in water or feces to 4.7% in animals [3]. This indicates a higher sequence abundance and diversity in animal than in environmental samples. In addition, the highest occurrence of *P. aeruginosa* has been found in sewage samples with 90% positive samples, followed by human feces with 11% positive samples [5].

A recent study revealed that the occurrence of *P. aeruginosa* is about seven times higher in a hydrocarbon- and pesticide-contaminated environment than in a noncontaminated environment [3]. In addition, samples derived from animals or humans or environmental samples associated with intense human activity showed increased *P. aeruginosa* contamination compared with environmental samples without human influence [3]. Close proximity between *P. aeruginosa* and human habitat may originate from the bacteria’s ability to metabolize oils, waste, and pesticide and their inherent resistance to heavy metals [6,7,8,9]. Such an environment is generally hostile to other bacteria and thus may provide a niche for *P. aeruginosa*. Overlapping of the two habitats provides an explanation for the increased infection risk and pathogenicity of *P. aeruginosa*. Further, the shared habitat of humans and *P. aeruginosa* imposes a constant selection and therefore contributes to the negative spiral of the multidrug resistance of this pathogen.

The risk of infection with *P. aeruginosa* is highest in patients with a compromised immune system or a chronic lung disease, patients on breathing machines or catheters, and those with infection of wounds from surgery and burns [10]. A risk factor assessment of *P. aeruginosa* bacteremia revealed the following conditions that facilitate an infection. Male intensive care patients who were previously or currently treated with antibiotics and had a urinary tract catheter or any postinstrumentation are at elevated risk of infection with *P. aeruginosa* [11]. In addition, a study investigating the risk factors for *P. aeruginosa* community-acquired pneumonia determined five risk factors associated with an infection [12]. These risk factors are previous infection with *P. aeruginosa*, tracheostomy, bronchiectasis, intensive respiratory or vasopressor support (IRVS), and very severe chronic obstructive pulmonary disease (COPD) [12]. A research performed in a hospital concluded that the risk factors for an infection depend on the contamination of sanitary facilities (including water), mechanical invasive ventilation, and inactive antibiotic prescription during the patient’s hospitalization [13,14]. Another study carried out in a hospital setting confirmed the previously mentioned risk factors (e.g., mechanical invasive ventilation, COPD, contamination of sanitary facilities (including water), and previous hospitalization) [15]. Moreover, it was revealed that older patients are at higher risk of infection [16]. In conclusion, the primary literature is in strong agreement regarding risk factors leading to infections with *P. aeruginosa*. These factors are IRVS, previous hospitalization, contamination of sanitary facilities (including water), and treatment with inactive antibiotics. Out of these four common risk factors, only two have the potential for improvement. The two factors that can be improved are prescription of inactive antibiotics and hygiene of sanitary facilities (including water). These two possibilities are also in agreement with the recommendations of the WHO [17].

“The virulence of a microorganism is defined by its ability to associate itself with a prospective host, to invade and multiply within that host, to produce local and/or systemic disease, and ultimately to overwhelm and even kill the host.” These dramatic words were used by Matthew Pollack to describe the virulence of *P. aeruginosa* in relation to the human host in 1984 [18]. According to this definition, the pathogenesis of *P. aeruginosa* is described with three steps. Step 1: bacterial attachment and colonization; step 2: local invasion; and step 3: disseminated systemic disease that may stop at any stage [18]. Tremendous efforts improved the understanding of host–pathogen interaction and slightly changed the second part or third step of the definition. The bacteria are multiplying within a host, but their “goal” is not to cause a disease or kill the host [19]. Therefore, the systemic disease is not a “goal” of the bacteria; rather, it is simply a result of numerous virulence factors maintained by *P. aeruginosa* combined with a compromised host immune system [20,21]. In general, virulence factors facilitate host invasion, evade host defenses, and thus cause a disease [19]. In *P. aeruginosa*, these virulence factors include adherence factors, invasion factors, endotoxins, exotoxins, and siderophores [22,23,24]. Out of these virulence factors, quorum sensing (QS), lipopolysaccharide (LPS), and type III secretion system (T3SS) have been determined as especially crucial in the pathogenicity of *P. aeruginosa* [21,25].

QS is described as cell–cell communication between bacteria to fulfill tasks as a colony by a coordinated response to environmental stimuli [25,26]. The underlying molecular mechanism of QS comprises three main pathways in *P. aeruginosa* [26]. These three pathways—las, rhl, and pqs—constitute a network that allows for cross-talk and thus results in fine-tuning of the response that is tailored to the environmental stimuli [22,27,28,29,30,31]. Apparently, the number of pathways constituting QS in *P. aeruginosa* have been heavily debated that peaked in a short comment clarifying that, indeed, QS is composed of three pathways in *P. aeruginosa* [30]. Understanding the detailed mechanisms and pathways of QS might provide potent target sites for antimicrobial agents against the pathogen. These targets are of tremendous interest because of their crucial role in cell–cell communication and the pathogenicity of *P. aeruginosa* [21,32,33]. A screening of 12 active antibiotics revealed that, indeed, three of them (azithromycin, ceftazidime, and ciprofloxacin) decrease the expression of QS-regulated virulence factors [34]. The reason for this decrease might be found in changes of the membrane permeability for certain QS factors [34]. Moreover, a structure-based virtual screening approach against the QS receptor lasR has resulted in five active compounds capable of inhibiting QS-regulated gene expression in *P. aeruginosa* [35]. Using such compounds in combination with classic antibiotics is a promising strategy to overcome the notoriously resistant pathogen *P. aeruginosa* [36]. Therefore, using signaling molecules of QS or a derivate thereof as Trojan horse to overcome multidrug resistant *P. aeruginosa* provides a powerful strategy in the fight against pathogenic bacteria.

LPSs are the most abundant surface-associated virulence factors composed of three domains: lipid A, core oligosaccharide, and distal O antigen [37,38,39]. These LPSs carry out a variety of functions in *P. aeruginosa*, ranging from a structural component of the outer membrane to a permeability barrier against small hydrophobic molecules and mediating bacterial–host interactions [38,40,41,42,43]. An immune response is triggered upon exposure of antigens to the host immune cells. In humans, LPSs are transported to immune cells via LPS-binding proteins. These LPSs are then transferred to CD14 located in the host immune cell membrane, where the LPSs are presented to toll-like receptor 4, eventually triggering an immune response [44]. In addition, exopolysaccharides, such as Psl, Pel, and extracellular DNA, are abundant components of the biofilm in *P. aeruginosa* [45,46]. These components adhere to each other and play an essential role in the highly complex biofilm formation, which is beneficial for growth and antibiotic tolerance [47,48,49]. *P. aeruginosa* expresses the soluble lectins lecA and lecB, both surface proteins capable of binding the exopolysaccharides [50]. These binding molecules have been both successfully targeted by glycopeptide dendrimers and monosaccharides, disrupting the biofilm formation [50,51,52,53,54]. Interestingly, it has been shown that the application of LPSs from *P. aeruginosa* stabilizes and increases the biofilm formation of other Enterobacteriaceae [55]. In conclusion, LPSs contribute to the defense and communication system, while exopolysaccharides contribute to the biofilm formation. Hence, both play a major role in the high virulence of *P. aeruginosa*.

T3SS is an injection apparatus used by *P. aeruginosa* to infuse effector proteins into host cells [56,57,58]. The large number of genes (about 36) involved in regulating and encoding for T3SS demonstrates the importance and effort carried out by *P. aeruginosa* to maintain this virulence factor [58]. Expression of these genes is regulated by various complex signaling pathways responding to extracellular and intracellular cues [59]. These signaling pathways are activated upon host cell contact, which ultimately leads to expression of T3SS and effectors [57,59]. The expressed macromolecular complex T3SS comprises four entities: a basal body residing in the cell membrane, an export apparatus providing energy and substrates, a needle filament, and a translocation pore formed within the host cell membrane [56]. Once this secretion system is activated by host cell interaction, only four effectors (ExoS, ExoT, ExoU, and ExoY) are transferred through the newly assembled pore. This is particularly interesting because T3SS of other pathogens transfer up to 25 effector proteins [58]. Thus, it is also true for the pathogenicity of bacteria that it is not always the quantity but the quality that makes the difference. Nevertheless, these four effectors potentially lead to collateral tissue damage, superinfection, bacteremia, and septic shock and reduce the oxygenation in infected lung tissue [58]. Given the severe effects of this virulence factor, recent advances in the discovery of new antimicrobials have targeted T3SS. These efforts have led to remarkable 12 new classes of small-molecule inhibitors and two types of antibodies against the T3SS of *P. aeruginosa* [60,61]. These promising results encourage us to further investigate the virulence factors and their mechanisms in *P. aeruginosa* to ultimately discover new and more potent antimicrobials for the fight against multidrug resistant pathogens.

Besides its harmful virulence factors, *P. aeruginosa* is a multidrug resistant pathogen capable of acquiring genes encoding resistance determinants in addition to its already highly developed intrinsic antibiotic resistance [62,63,64]. Detected resistances of antibiotic classes in *P. aeruginosa* include β-lactams, fluoroquinolones, and aminoglycosides [65]. These resistances are achieved by expression of different types of β-lactamases and aminoglycoside-modifying enzymes; loss of oprD, an outer membrane porin; mutations in gyrA, a topoisomerase; and overexpression of various multidrug efflux pumps [62,65]. The versatile resistance mechanisms of *P. aeruginosa* cause tremendous challenges to the therapy of infected patients. These include the intrinsic resistance and, in particular, the ability to acquire resistance during antimicrobial treatment [66]. Thus, serious infections are usually treated with a strain-specific combination of antibiotic classes (e.g., β-lactam and aminoglycoside) that results in a synergistic effect [66]. Unfortunately, emerging resistance even against last-resort antibiotics requires novel antimicrobials and targets to successfully combat multidrug resistant *P. aeruginosa*. Consequently, the development of β-lactamase inhibitors is one promising approach to overcome the widespread β-lactam resistance [67,68]. Structural studies of penicillin-binding protein 3 have revealed spatial rearrangements of secondary motifs upon inhibitor binding, thus providing novel insights to improve existing β-lactamase inhibitors [69]. Another approach takes advantage of *P. aeruginosa*’s own virulence factor. In that respect, a γ-lactam was conjugated to a siderophore moiety, taking advantage of the intrinsic siderophore uptake mechanism of *P. aeruginosa* [70]. These creative and effective strategies in combating emerging broad-spectrum resistance demonstrate that only continuing research and effort lead to novel antimicrobials eventually overcoming the multidrug resistance of *P. aeruginosa*.

In conclusion, *P. aeruginosa* is a skilled survivor in a moist environment that is adverse to many other bacteria, thus cultivating its own niche. Because this niche is tightly connected to and influenced by humans, chances of infections are exceptionally high. Factors that increase the risks of infection are purity of water and sanitary installations, a weakened immune system, previous hospitalization with IRVS, and misuse of antimicrobials. Once a host is infected, *P. aeruginosa* maintains several highly regulated virulence factors, such as QS, LPS, and T3SS, contributing to the severe pathogenesis, eventually leading to the outbreak of a disease or worse. During this pathogenesis, *P. aeruginosa* is protected by its intrinsic resistance, adaptive resistance, and ability to acquire additional resistance determinants. Given the high infectivity and strong protection, it is of utmost importance to understand the entire homeostasis and pathways found in *P. aeruginosa*. This understanding might reveal a weak spot to successfully target the multidrug resistant pathogen. The succeeding paragraphs focus on recent advances in the understanding of copper homeostasis in *P. aeruginosa*. Further, we emphasize the use of heavy metals as antimicrobial agents also in combination with other drugs and antibiotics to overcome the emerging broad-spectrum resistance of *P. aeruginosa*.

## 2. Copper Import

Copper is an essential nutrient; concurrently, high copper concentrations are deleterious to the bacteria [71]. Therefore, *P. aeruginosa* develops and maintains a complex copper homeostasis to preserve the thin line between essential copper concentrations and excess of copper (Figure 1). It is a topic of current research to verify and understand the detailed mechanisms responsible for copper uptake in bacteria [72]. To date, there is a common understanding that porins, such as oprC, are responsible for the majority of copper detected in bacteria [72,73,74]. Very recent advances suggest that the TonB-dependent transporter oprC can bind to Cu(I) and mediate the import and oxidation to Cu(II) (6). Still, the transfer from the periplasm to the cytoplasm remains enigmatic. Only sparse data are available about the inner membrane transport. It has been suggested that copABCD is responsible for elevated copper concentrations found in *Pseudomonas syringae* [75]. CcoA is another copper import protein and belongs to the major facilitator superfamily (MFS) transporter in *Rhodobacter capsulatus* [76]. These MFS transporters are known to transport various substrate classes across the membrane [77,78,79]. Unfortunately, we were not able to find a ccoA homolog protein sequence or a homolog nucleotide sequence with more than 50% identity in *P. aeruginosa*. Absence of a close homolog indicates a lack of a dedicated ccoA copper import protein in *P. aeruginosa*. Aside from porins, it is reported that the siderophores pyoverdine (PVD) and pyochelin (PCH) chelate copper [80,81]. However, experimental evidence has shown that solely iron is efficiently transported into the bacteria via the two siderophores [82]. Still, these findings do not exclude the import of copper by siderophores and indicate a dependence on copper concentration for the import facilitated by siderophores [83,84]. Altogether, oprC has been shown to facilitate copper import into *P. aeruginosa* [6,73]. Aside from these porins, copABCD and siderophores might contribute to the copper import, albeit in a concentration-dependent manner. Additional research is urgently required to fully understand the import mechanisms and the interplay between these proteins to maintain the essential copper concentration in *P. aeruginosa*.

## 3. Copper Defense Mechanisms in *P. aeruginosa*

Human pathogens develop numerous defense strategies and resistance mechanisms against the deleterious effects of high copper concentrations. Here, we focus on the proteins involved in the copper defense strategies of *P. aeruginosa* and compare these with the ones in *E. coli*. Known proteins involved in copper metal homeostasis in *P. aeruginosa* and *E. coli* are depicted in Figure 1 and listed in Table 1 [85]. Homologous proteins found in *P. aeruginosa* are listed in the first lane. Proteins are grouped according to transcription factor (TF), metallothionein (MT), cytoplasmic copper chaperone (CYTO-C), copper-sensing two-component systems (2CS), P-type copper ATPase (P-type), resistance-nodulation-division-type transmembrane efflux pump (RND), periplasmic copper chaperone (Peri C), multicopper oxidase (MCO), and siderophores (Sidero) (Table 1), according to Alex G. Dalecki et al. [85]. Major differences of proteins involved in copper homeostasis between the two species have been found in MT, CYTO-C, RND, and Sidero. *E. coli* lacks MT, and it does not express any cytoplasmic copper chaperones. However, it contains a dedicated cusCBA system for copper export. Moreover, the types of siderophores involved in copper homeostasis are reduced compared with *P. aeruginosa* (Table 1).

**Table 1 ijms-22-02050-t001:** Proteins involved in copper homeostasis in *P. aeruginosa* and *E. coli*. A reference indicates experimental evidence of copper homeostasis. * indicates a potential role in copper homeostasis based on the homology alignment or structural prediction of proteins found in other pseudomonad strains and bacteria. TF: transcription factor; MT: metallothionein; CYTO-C: cytoplasmic copper chaperone; 2CS: copper-sensing two-component systems; P-type: P-type copper ATPase; RND: resistance-nodulation-division-type transmembrane efflux pump; Peri C: periplasmic copper chaperone; MCO: multicopper oxidase; Sidero: siderophores; N/A: not applicable. Table adapted from [85].

	TF	MT	CYTO-C	2CS	P-Type	RND	Peri C	MCO	Sidero	Others
*P. aeruginosa*	cueR [86]	MT * [87]	copZ1 copZ2 [86,88]	copRS [89]	copA1 copA2 [90]	czcCBA *	ptrA [91] azurin [92]	pcoA [93]	PVD PCH [82,83,84]	pcoB * [94]
*E. coli*	cueR [95]	N/A	N/A	pcoRS [96] cusRS [97]	copA [93]	cusCFBA [98]	pcoE [99] pcoC [100] cusF [98]	pcoA [101] cueO [102]	Ybt [103]	cut [95] pcoB [104] pcoD [104] porins

Once copper ions reach the cytoplasm or periplasm of *P. aeruginosa*, the copper defense cascade is triggered (Figure 1 and Table 1). This defense strategy can be divided into three stages: (1) scavenging, (2) first responder, and (3) second responder.

First, copper is scavenged by copZ1 (PA3520) and copZ2 (PA3574.1), followed by the activation of the first responder copper efflux regulator (cueR, PA4778) in the cytoplasm [105]. Copper scavenging is carried out by the two copper chaperones copZ1 and copZ2 (Figure 1) [105]. The two copper-binding proteins show $K_{D}=\;4\;\cdot \;10^{-15}$ and $K_{D}=\;8\;\cdot \;10^{-17}$, respectively. Supposedly, copZ1 transports Cu(I) to the first responder cueR, and copZ2 transfers Cu(I) to copA1 [105]. CopA1 (PA3920) and copA2 (PA1549) are both copper transport ATPases known to transfer copper from the cytoplasm to the periplasm [106]. Based on homology search, there is a third Zn transport ATPase with the locus tag PA3690 (see Figure 1) [88]. Whether this Zn P-type ATPase contributes to the copper export remains to be answered. Even though PA3690 knockout mutants show no sensitivity to copper exposure, the expression profiles during copper shock treatment indicate a regulation by the first responder cueR [88,107]. Whether the transcription of copA2 and the third Zn ATPase are also regulated by cueR or through other transcription factors is a topic of current research.

Second, cueR is the first responder and a cytoplasmic transcription factor that belongs to the family of mercury resistance operon regulatory protein (merR) transcription factors [108,109,110,111]. Both organisms *P. aeruginosa* and *E. coli* own a cueR transcription factor that is readily activated by miniscule copper concentrations [109]. It has been shown that cueR in *E. coli* has an exceptionally high Cu(I) affinity of $K_{D}\;=\;10^{-21}\;M\;\mathrm{to}\;3.25\;\cdot \;10^{-19}\;M$ [112,113]. Interestingly, the KD of *P. aeruginosa* cueR is about five to three orders of magnitude higher, $K_{D}\;=\;2.5\;\cdot \;10^{-16}\;M$, and thus displays lower copper affinity than cueR from *E. coli* [105]. It has been demonstrated that the interaction between holo-copZ1 and cueR facilitates its activation, whereas copZ2 accelerates copper sequestration to the periplasm through copA1 (PA3920) (Figure 1) [104]. The activation of cueR upon the binding of Cu(I) increases the transcription of the following proteins: copZ1/2, mexPQ-opmE (PA3521-3523), copA1 (PA3920), and PA3515-3519 (Table 2) [86,88,107]. These genes display a fast increase in transcription within an hour upon copper treatment (copper shock), thus the term first responder. Aside from the above-mentioned proteins, it has recently been reported that azurin (PA4922), oprC (PA3790), and the type VI secretion system (H2-T6SS) (PA1656–PA1659) might be regulated by cueR [114]. Unfortunately, these results have not been confirmed with either proteomic profiling or transcription analysis after copper treatment [107,115,116]. Interestingly, aside from the hypothetical proteins (Table 2), mexPQ-opmE is not directly involved in copper homeostasis. Overexpression experiments with mexPQ-opmE in *P. aeruginosa* have shown increased resistance to macrolides, fluoroquinolones, and other drugs [117]. MexPQ-opmE belongs to the family of resistance-nodulation-cell-division (RND)-type multidrug efflux pumps, and its transcription is regulated by cueR [86]. Thus providing a link between heavy metal homeostasis and multidrug resistance [117]. This link between heavy metal homeostasis and multidrug efflux pump is of great evolutionary advantage. Since Cu(I) is mainly found in an environment with human influence, it is very likely that antibiotics occur in conjunction with elevated environmental Cu(I) concentrations [3,118]. Therefore, the transcription regulation of mexPQ-opmE by cueR may provide an evolutionary advantage in habitats shared with humans.

Third, copRS (PA2809/PA2810) is the second responder, in case copper concentrations remain elevated within the periplasm (copper adapted). CopS exhibits a high affinity for Cu(I) and Cu(II) with a dissociation constant of $K_{D}\;=\;3\;\cdot \;10^{-14}\;M$ [119]. The two-component regulatory system is activated to increase the transcription of genes involved in copper sequestration and oxidation to ultimately reduce Cu(I) concentrations within the cell (Figure 1) [89]. Genes regulated by copRS are listed in Table 2. Transcription profiles of these genes after copper shock have displayed a delayed and prolonged increase even six hours after copper treatment [88,107]. Therefore, the term second responder indicates copper-induced transcription regulation over an extended period of time by copRS (copper adapted). There is controversial literature on how copR is regulated by copS. The prototypical mechanism indicates that upon copper binding, copS is autophosphorylated. This phosphorylation is subsequently transferred to copR, which allows binding and transcription of specific DNA regulons [120,121]. A recent activation mechanism proposed a persisting phosphorylated copR, which is dephosphorylated by apo-copS [119]. Quantitative levels of copRS phosphorylation, phosphorylation transfer, and whether there is a cross-talk between copRS and czcRS remain to be answered. A potential cross-talk between the two systems would explain why copR or czcR is active even in the absence of copS or czcS, respectively [122].

**Table 2 ijms-22-02050-t002:** Copper-dependent transcription regulators cueR and copRS, including regulated genes with locus tag and protein name where available. * indirect upregulation of czcCBA through overexpression of czcR [123].

Transcription Regulator	Locus Tag	Regulated Proteins	Reference
cueR (first responder) *PA4778*	PA3515–PA3519	Five hypothetical proteins potentially involved in glycolysis and fatty acid metabolism	[86,88]
	PA3520	copZ1	[86,105]
	PA3521–PA3523	mexPQ-opmE	[86,88,117]
	PA3574.1	copZ2	[86,88,123]
	PA3920	copA1	[86,105]
copRS (second responder) *PA2809/PA2810*	PA2065	pcoA	[88,107]
	PA2064	pcoB	[88,107]
	* PA2520–PA2522	czcCBA (in conjunction with cadA activity)	[88,122,124,125]
	PA2523	czcR	[88,122,126]
	PA2524	czcS	[88,122,126]
	PA2806	Hypothetical protein: potentially a NADPH-dependent reductase	[107]
	PA2807	Hypothetical protein: azurin/plastocyanin family	[88,107]
	PA2808	ptrA	[88,91]
	PA0958	oprD (downregulation with cofactor: Hfq)	[88,127]

PcoA (PA2065) and pcoB (PA2064) are both outer transmembrane-spanning proteins responsible for reducing copper concentrations in the periplasm by excretion and oxidation, respectively (Figure 1) [72,128]. It is thought that these two membrane proteins are in close interaction and might orchestrate the export and redox reaction of copper [128,129,130]. These findings are based on close homologs observed in *E. coli*; moreover, the overlapping start codon of pcoB downstream of pcoA indicates a simultaneous transcription of these genes as found in *P. syringae* [101,131]. Furthermore, homology search indicates that pcoA catalyzes the oxidation of Fe(II) to Fe(III), comparable to the oxidoreductase Fet3p found in *Saccharomyces cerevisiae* [93,101]. Additionally, knockout studies have demonstrated that pcoA is required for iron uptake in *P. aeruginosa* and is indeed a ferroxidase [132]. However, how the two proteins pcoA and pcoB interact with each other, including the detailed mechanisms of the copper redox reaction, remains elusive.

CzcCBA (PA2520–PA2522) belongs to the family of heavy metal efflux (HME) RND pumps and is known to facilitate the export of Co(II), Zn(II), and Cd(II) (czc) [125]. Figure 2A shows a schematic representation of an RND pump consisting of three components: outer membrane factor (OMF) (Figure 2C), membrane fusion protein (MFP) (Figure 2D), and cytoplasmic membrane transporter (CMT) (Figure 2B). Further, a phylogenetic analysis was performed on RND sequences for each of the RND components. The RND components were selected from known RND efflux pumps listed by Philip D. Lister et al. [65]. In addition, czcCBA was included to verify the distance to RND efflux pumps with known substrates. It becomes apparent that mexPQ-opmE (PA3521–PA3523) is unlikely to be responsible for copper efflux as proposed elsewhere [107]. A comparison of the distances between *E. coli* cusCFBA and RND components of *P. aeruginosa* does not indicate a close relation between mexPQ-opmE and cusCFBA (Figure 2). In other words, the phylogenetic analysis indicates a closer relationship between czcCBA of *P. aeruginosa* and cusCBA of *E. coli* than mexPQ-opmE, even though mexPQ-opmE is regulated by the copper-sensitive first responder cueR [117]. Based on the phylogenetic analysis, it is possible that czcCBA also participates in the export of copper (Figure 2). Given that czcCBA transcription is indirectly regulated by the copper-sensitive second responder copRS, including the activity of cadA, it is likely that czcCBA contributes marginally to copper export [122,124]. Moreover, the deletion of czcA in *P. aeruginosa* has shown an increase in zinc sensitivity but no change in copper sensitivity, thus confirming only a marginal role in the copper homeostasis of *P. aeruginosa* [88]. Whether *P. aeruginosa* maintains a dedicated RND efflux pump for copper export or whether czcCBA facilitates copper export in a negligible amount along with the other mechanisms described above is a topic of ongoing research.

CzcRS (PA2523 and PA2524) is a two-component system, and its transcription is upregulated mainly upon elevated zinc concentrations, but also by copRS and copper exposure [122,126]. Most importantly, czcRS downregulates the expression of the oprD porin (PA0958) in the presence of cofactor Hfq [127]. This downregulation results in carbapenem resistance because of reduced antibiotic import [126]. Thus, there is an indirect link between copper and carbapenem resistance through the activation of copRS and czcRS and the subsequent reduced expression of oprD [122,126]. Obviously, the cross-talk between the two heavy metal regulation systems czcRS and copRS increases the complexity of copper and zinc resistance mechanisms in bacteria. Thus, only a comprehensive view that includes both systems will provide sufficient understanding of how *P. aeruginosa* regulates the delicate levels of copper and other heavy metals. A recent proteomic analysis of copper stress in *P. aeruginosa* was not able to support this indirect link between copper stress and reduced oprD expression [115]. Hence, raising the fundamental question of reliability and coherence between transcriptome analyses and proteome profiling in general [116].

PtrA (PA2808) is a copper-binding protein located in the periplasm, therefore contributes to the copper scavenging and copper tolerance of *P. aeruginosa*, such as azurin [91,92]. However, it is claimed that ptrA also has an impact on the transcription regulation of T3SS [135]. This controversial finding provides the basis for the disputable name *Pseudomonas* type III repressor A (PtrA). For this reason, further research is urgently required to precisely clarify the protein’s function and adjust the nomenclature.

Altogether, the copper defensive strategy in *P. aeruginosa* can be divided into three stages: scavenging, first responder, and second responder. Scavenging of free copper within the cytoplasm is facilitated by the copper chaperones copZ1 and copZ2, which leads to subsequent activation of the first responder cueR [105,110]. Additionally, prolonged elevated copper concentrations within the periplasm lead to activation of the second responder, copRS [119,136]. Both copper-binding transcription regulators facilitate activation or repression of the gene transcription listed in Table 2. These proteins enable copper export, copper oxidation, contribute to antibiotic resistance, and therefore, lower the elevated and deleterious Cu(I) levels within the cell and grant survival in a toxic niche (Table 1, Figure 1).

## 4. Structural Insights into Proteins Involved in Copper Homeostasis

CueR belongs to the merR transcription regulator family and is the first responder in case of excess copper. Phylogenetic analysis of merR in pathogenic bacteria highlights versatility, importance, and prominence of this merR transcription regulator throughout pathogen species (Figure 3B). Strikingly, cueR of *P. aeruginosa* is capable of binding five different DNA sequences, whereas in *E. coli*, cueR is limited to only two DNA binding sequences [86]. Lately, substantial progress in understanding the dynamics of *E. coli* cueR was achieved by EPR and cryo-EM [109,110,111,137]. These techniques allow measurement of the transcription activation mechanism induced upon Cu(I) binding of cueR. It has been demonstrated that cueR in *E. coli* induces a kink within the DNA at the −35 element, allowing the transcription factors σ2, σ4, and RNA polymerase to bind at the opposite site of the DNA [109,111]. These findings confirm the observations that the transition and dynamics of cueR from a repressed to an active state are independent of the presence of transcription factors or RNA polymerase [110]. Only with protein crystallography and EPR it was possible to describe the apo or repressed state of *E. coli* cueR [109,138]. Structural comparisons of different bound metal ions have revealed that there are only minor differences between the structures (e.g., RMSD = 0.2 Å) [112]. Further, it was shown with EPR that binding DNA reduces the distance between the binding sites for about 1 Å (Figure 3A) [109]. However, the largest change between the two DNA-binding domains in cueR occurs upon binding of the metal ion. EPR data indicate a reduction of about 17 Å upon Cu(I) binding, whereas X-ray crystallography only shows a change of about 11 Å between the two DNA binding sites (Figure 3A) [109,138]. The latest cryo-EM structure reveals a change of 14 Å between repressed and active state [111]. The difference of the distances is calculated between the amino acid residue glycine 11 of the repressed apo-cueR (PDB-ID: 4wls) and the corresponding active state (PDB-ID: 4wlw, 6ldi). A difference of 3 Å between EPR measurement and the structural data derived from X-ray crystallography and cryo-EM indicates a further change upon RNA polymerase or transcription factor binding. The interaction between the RNA polymerase sigma factor (rpoD) and the DNA on the opposite site of cueR might provide an explanation for the difference between the active states of cueR. Hence, the binding of rpoD to the DNA might provide a slight relaxation of the DNA kink induced by cueR. Despite remarkable progress in the past years, more questions were raised to fully understand the transcription activation induced by cueR and RNA polymerase, particularly the role of the different DNA binding sequences found in *P. aeruginosa*, remain unanswered.

CusCFBA belongs to the RND superfamily. It is widely accepted that this protein family comprises the most potent efflux machineries found in bacteria [129,130]. Consequently, this superfamily contributes immensely to antibiotic resistance found in Gram-negative bacteria [139,140]. In *E. coli*, cusCFBA is responsible for the heavy metal efflux of Cu(I) and Ag(I) ions [88,131]. So far, no experimental evidence was provided that the same system is found in *P. aeruginosa*. Phylogenetic analysis carried out in the section above indicates that there is no close homolog of cusCFBA in *P. aeruginosa* except czcCBA. However, the deletion of czcA in *P. aeruginosa* displays no change in copper sensitivity, indicating that czcCBA has no or only a marginal role in the copper homeostasis of *P. aeruginosa* [97]. Because transcription of mexPQ-opmE (PA3521–PA3523) is tightly regulated by cueR, it has been proposed that this RND pump is a cusCBA homolog responsible for the copper efflux in *P. aeruginosa* [98]. Here, we predicted two models of the CMT mexQ and performed structural analysis to verify the presence of crucial methionine residues in cusA required for heavy metal transport of this type of RND efflux pump (Figure 4) [88]. Further, other RND heavy metal pumps facilitate the ion efflux by charged residues instead of methionine residues. Therefore, these charged residues were considered while assessing the role of mexPQ-opmE in heavy metal export. Unfortunately, we were not able to locate either the methionine residues or the charged residues required for Cu(I) binding in the binding site of the mexQ models (Figure 4A). Hence, it is very unlikely that mexPQ-opmE acts as a copper efflux pump in *P. aeruginosa,* based on the model prediction of mexQ (Figure 4B,C). Recently, it was published that the cusCFBA system is lost and replaced by cueP and scsABCD genes containing thiol oxidoreductases and putative cuproproteins in the pathogen *Salmonella* [141], indicating that the cusCBA system is the outdated copper efflux system that is exchanged by cueP and scsABCD through evolution. Taken together, phylogenetic analysis has indicated that there is no close relationship between mexPQ-opmE and cusCBA (Figure 2). Further, models of mexQ based on cusA or multidrug efflux pumps did not provide a methionine-rich copper binding site, nor were charged residues present to complex Cu(I) (Figure 4A–C). Finally, there is literature demonstrating that mexPQ-opmE is indeed a multidrug efflux pump [117]. Therefore, it is highly unlikely that mexPQ-opmE is involved in copper or any other heavy metal export.

## 5. Copper as Antimicrobial

The effect of antimicrobial metals has been used for millennia, dating back as far as 2400 BC, where Egyptian surgical tools were made out of copper [144]. During the same time, copper was also used to sterilize water and wounds in ancient Egypt [145]. Later on, in the 18th century, copper found intensive application in agriculture and clinical use due to its antifungal and antimicrobial properties [145,146]. The pinnacle of copper as antimicrobial alloy was the approval by the U.S. Environmental Protection Agency (EPA). That acknowledged that copper, brass, and bronze are capable of killing 99.9% of Gram-negative and Gram-positive bacteria. Therefore, copper is the first solid surface material receiving an EPA registration back in 2008 [145]. Another remarkable effect of copper is its antiviral property. This discovery dates back to 1964, where Cu(II) was shown to inactivate bacteriophages [147]. Since then, many more types of viruses have revealed their susceptibility to copper (e.g., single- or double-stranded DNA or RNA enveloped or nonenveloped viruses), one of which being HIV [145,148]. Additionally, copper used in hospital settings has shown that bed rails made out of copper reduce the bacterial burden and healthcare-acquired infections because of its continuous antimicrobial activity [149,150]. Moreover, linen with copper fabrics or impregnated with copper oxide has shown efficient reduction of healthcare-associated infections [151,152]. The antimicrobial, antifungal, and antiviral activity or, in short, the biocidal activity of copper is based on two mechanisms [153]. These two mechanisms are membrane damage and oxidative stress, which eventually results in the destruction of genetic material [153,154,155]. The ability of copper to damage both DNA and RNA through the formation of reactive oxygen species also explains its antiviral activity [146,156]. The history of copper throughout the ages emphasizes the paramount advantages of the long-lasting use of copper and its alloys as antimicrobial agents. The nearly endless biomedical potential of copper renders this metal a superior material for use in heavily frequented public places, such as hospitals, schools, airports, train stations, and public transportation.

## 6. Copper and Antibiotics—The Power Couple

Copper has been used as antimicrobial by mankind for millennia. Surprisingly, the use of copper as part of the defense strategy against an invading species dates even further back in time. Recently, it was revealed that cells of the innate immune system take advantage of the antimicrobial effects of copper [157,158]. It is possible to show that macrophages and other cells increase the ATP7A-mediated copper trafficking into phagolysosomes with engulfed bacteria by the overexpression of the ATP7A P-type ATPase [159,160]. Lasting and specific use of copper by immune cells demonstrates that copper provides a persistent evolutionary advantage in the fight against multidrug resistant pathogens. Thus, using copper or other heavy metals in a combinatorial drug treatment might enhance the efficacy and potency of current and future antibiotic treatments. In the following, we list known drugs that were used in conjunction with copper to combat bacteria based on R. Poole et al. [85]. Table 3 lists drug names or antibiotic types, including the mode of action of the cooperative effect between copper and drug (copperaction).

In general, it was noted that all compounds listed in Table 3 form a complex with copper. Hence, the listed drugs in Table 3 are capable of delivering copper into the pathogen cell as copper ionophore. The delivery of both the drug and copper will stress bacteria in two ways. First, it will increase the amount of reactive oxygen species within the cell. Second, the antibiotic or drug will challenge the bacteria through the antibiotic’s specific mode of action. The copperaction of ditiocarb has been heavily debated and assigned to the copper toxicity derived from macrophages [85]. Here we disagree with this explanation and propose that ditiocarb might interact with csoR or ricR and bind to the cysteine residues found at the metal binding site. Thus, ditiocarb might increase the efficacy of copper by delivering copper and act as a suicide inhibitor by potentially linking the two cysteine residues at the metal binding site of csoR or ricR. 8-Hydroxyquinoline (8HQ) is a bidentate metal chelator capable of forming a Cu(II) complex with a different stoichiometry (e.g., Cu(II)–8HQ; 1:1 and 1:2 complex) [172]. The Cu(II) complex acts as an ionophore and increases the cell-associated labile copper ions, ultimately facilitating the destruction of *Mycobacterium tuberculosis* in primary macrophages. Additionally, it was shown that only the 1:1 stoichiometry exhibits antibacterial activity in *M. tuberculosis* [162]. Thiosemicarbazones are tetradentate chelators capable of complexing Cu(II) and act as an ionophore similar to 8HQ but contain rotatable bonds, increasing the flexibility of the scaffold [161,173]. Thiosemicarbazones themselves are thought to impair respiratory enzymes, such as dehydrogenases [163,174]. In addition, it has been demonstrated that this class of molecules also has activity independent of the redox cycle. Thus, it might be possible that thiosemicarbazones also impair copper-binding proteins via cysteine residues and their two sulfur groups as described above. Phenanthrolines are bidental chelators known for their membrane permeability. In addition to the copper transport, phenanthrolines are capable of DNA strand scission and inhibition of RNA polymerase, resulting in their enhanced antimicrobial activity [166]. Pyrithione is a bidental chelator similar to 8HQ but with a four-times-higher potency than 8HQ [175]. Its activity is solely based on its activity as a copper ionophore without any biocidal activity itself [167]. The classic antibiotics tetracycline, fluoroquinolone, and aminoglycosides are bidental chelators showing copper ionophore activity in a 1:1 or 1:2 (metal–ligand) complex, in addition to their antimicrobial mechanism of action [85]. In summary, combination of the above-mentioned antibiotics with copper displays a prototypical use of combinatorial drug treatment against human pathogens, such as *P. aeruginosa*. A more detailed explanation of the synergistic mode of action can be found in the book of R. Poole et al. [85].

The circumstance that copper has been selected by evolution over millennia and is still employed by the innate immune system indicates its superiority in antimicrobial efficiency and potency. Thus, future development of effective antibiotics has to consider the impact of copper during the innate immune response and take advantage of the unique copper homeostasis found in bacteria. Combinatorial drug treatment is a common strategy in the fight against fast-evolving cancer [176]. Because bacteria are also fast evolving, it seems evident that a combinatorial drug treatment that additionally targets the copper homeostasis provides a promising strategy in the fight against multidrug resistant pathogens [177].

## 7. Conclusions

Bacteria are an integral and essential part of nature’s microbiome. Consequently, there is a constant threat to humanity originating from pathogenic bacteria. More specifically, the human habitat and humans themselves provide and nurture the niche for *P. aeruginosa*, resulting in a never-ending competition. During this competition, which has lasted already for millennia, evolution has selected the antimicrobial effects of copper as the primary defense system against bacteria. It was shown that the innate immune system takes advantage of the well-established antimicrobial effects of copper. Thus, understanding the detailed mechanisms of copper homeostasis in *P. aeruginosa* will provide an advantage in the never-ending competition between humanity and multidrug resistant pathogens.

Here we divide *P. aeruginosa*’s reaction to elevated copper levels into three stages, resembling those of public safety organizations. Stage 1 is copper scavenging by copper-binding proteins, followed by the second stage, activation of the first responder and transcription regulator cueR. The third stage is carried out by the transcription regulator and second responder copRS, which regulates prolonged elevated copper levels. Each of these three stages provides itself numerous target sites to sabotage the defense strategy and increase the effects of antimicrobial copper derived from the innate immune system. Redundancy of some of these proteins within the copper homeostasis indicates the importance of regulation and fine-tuning of copper concentrations within the bacteria. Because of functional redundancy, only a combinatorial drug regimen targeting several proteins in parallel will provide the desired antimicrobial efficacy. Moreover, trivial solutions, such as fabrics or surfaces made out of copper or copper alloy in public areas, will provide a highly efficient and simple protection from bacteria, viruses, and fungi.

Altogether, we highlighted controversies in the nomenclature and function assigned to proteins regulated by cueR. The genome nomenclature provided by Winsor et al. would reduce such misunderstandings in the future [178]. We showed with phylogenetic analysis and structural modeling that mexPQ-opmE is unlikely to be involved in copper homeostasis but more likely to act as a multidrug efflux pump [117]. Moreover, cueR induces conformational changes in the DNA to facilitate binding of the RNA polymerase and transcription factors [109,111]. Despite tremendous advances in the field of structural biology, there are still open questions regarding detailed mechanisms and interactions between transcription factors, cueR, and DNA, leaving us with more research ahead until we fully understand the organization, mechanisms, and regulation of copper homeostasis in the human pathogen *P. aeruginosa*.

## Figures and Tables

**Figure 1 ijms-22-02050-f001:**
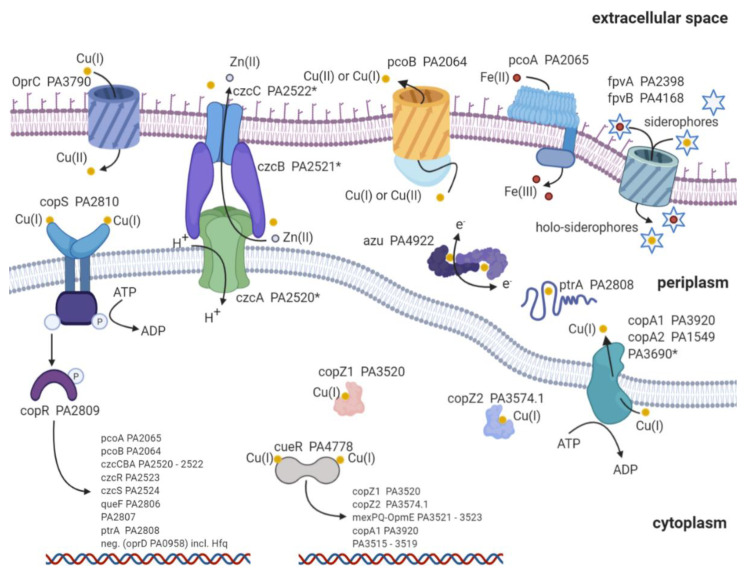
Proteins involved in the copper homeostasis of *P. aeruginosa* (Table 1). The proteins are depicted in a schematic representation. Copper is colored yellow, zinc grey, and iron red. The proteins are labeled with their name and locus tag. Where no name was available, only the locus tag is used. Genes regulated by the two transcription factors cueR and copRS are listed above the DNA. An extensive list of these proteins is found in Table 2. * indicates a potential role in copper homeostasis based on the homology alignment or structural prediction of proteins found in other pseudomonads and bacteria.

**Figure 2 ijms-22-02050-f002:**
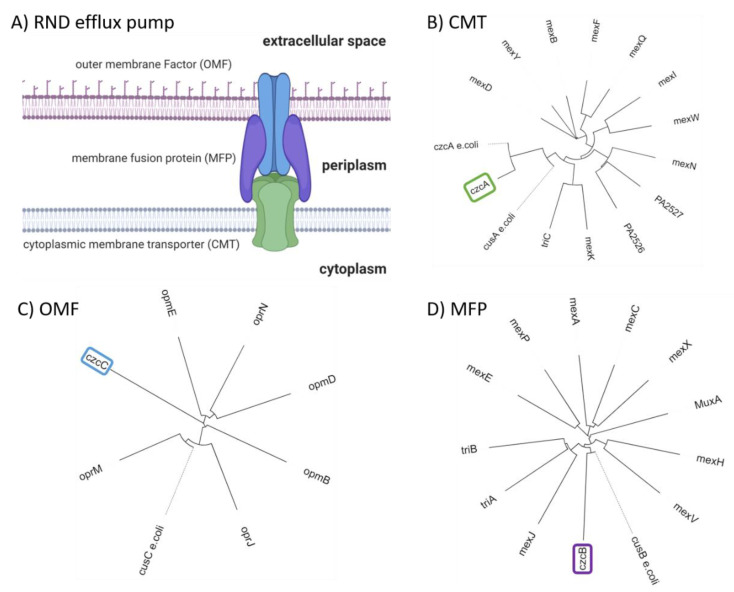
**(****A**) Schematic representation of an RND efflux pump as a tripartite system consisting of cytoplasmic membrane transporter (CMT), outer membrane factor (OMF), and membrane fusion protein (MFP). (**B**) Phylogenetic tree of *P. aeruginosa* CMT listed in [65] and *E. coli* cusA/czcA depicted with dashed line. (**C**) Phylogenetic tree of *P. aeruginosa* OMF listed in [65] and *E. coli* cusC depicted with dashed lines; no close homolog was found for czcC in *E. coli*. (**D**) Phylogenetic tree of *P. aeruginosa* MFP listed in [65] and *E. coli* cusB depicted with dashed line; no close homolog was found for czcB in *E. coli*. Calculated with Clustal Omega and illustrated with iTOL [133,134].

**Figure 3 ijms-22-02050-f003:**
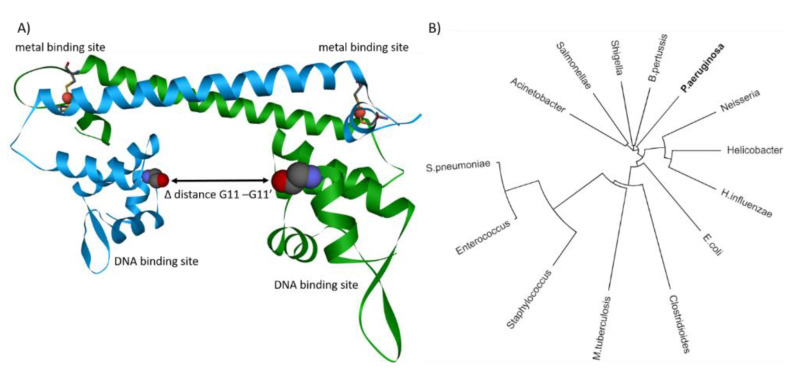
(**A**) CueR dimer with complexed Cu(I). The two metal binding sites are on top composed of two cysteine residues. The DNA binding sites are at the bottom acting as a hinge. Upon copper binding, the distance between the two hinges is reduced, resembling a clamp, which closes upon copper binding. The difference of the distance between active and repressed state was calculated between the two G11 residues. (**B**) Phylogenetic tree of *P. aeruginosa* cueR homologs and merR family members found in pathogenic bacteria. Calculated with Clustal Omega and illustrated with iTOL [133,134].

**Figure 4 ijms-22-02050-f004:**
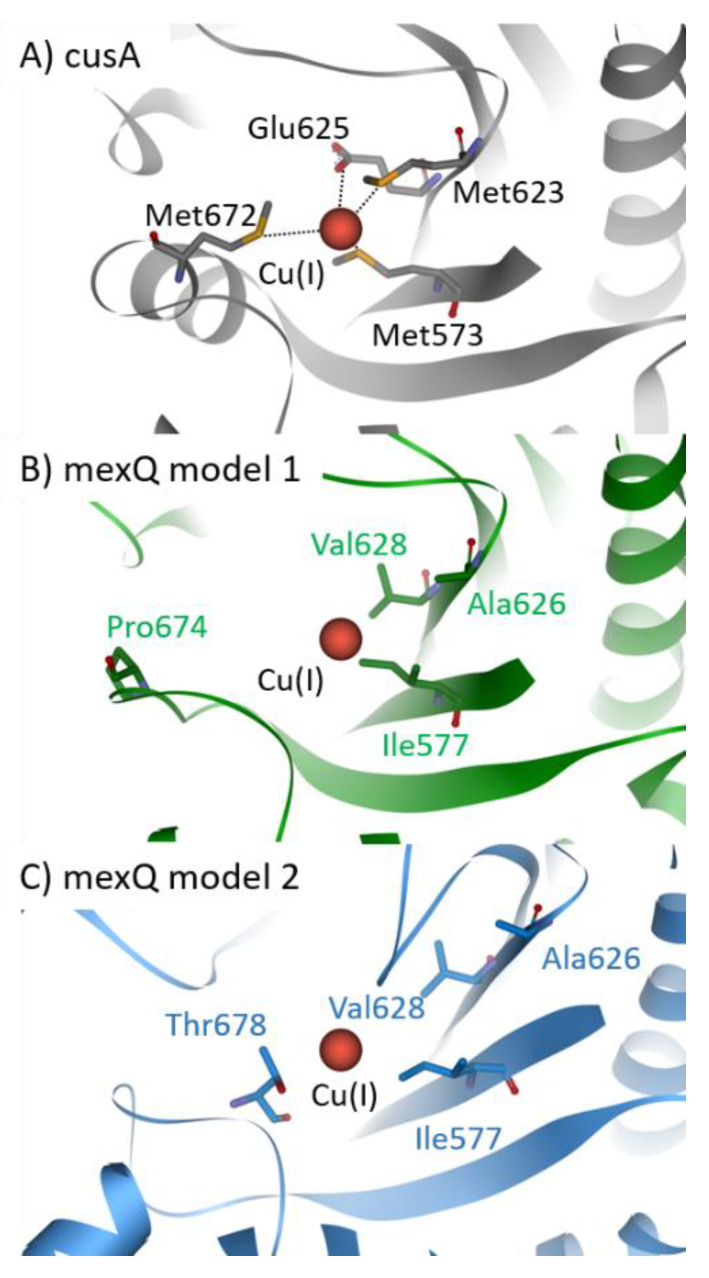
Copper binding site of cusA (PDB-ID: 3T53) (**A**). Two models of mexQ were predicted based on the structure of cusA (PDB-ID: 3T53) in green (**B**) and based on the structures of the top four multidrug efflux pumps (PDB-ID: 5T0O, 4DX5, 6VKT, and 6OWS) in blue (**C**) [142,143]. Cu(I) is complexed by three methionine residues (Met573, Met623, and Met672), including Glu625 in *E. coli* cusA, illustrated in the stick representation and colored according to the elements. Corresponding residues of the mexQ models were depicted in the stick representation and colored according to the parent protein model.

**Table 3 ijms-22-02050-t003:** Drugs that demonstrated a synergistic antimicrobial action in cooperation with copper (copperaction).

Antibiotic	Mode of Copperaction	Reference
Ditiocarb (Disulfiram)	Copper complex, bypasses the copper homeostatic machinery in *Mycobacterium tuberculosis*	[161]
8-Hydroxyquinoline	Copper complex and ionophore, thus facilitates the transfer of copper across hydrophobic membranes.	[162]
Thiosemicarbazones	Copper complex and ionophore, target NADPH dehydrogenases	[163]
Phenanthroline	Copper complex with nuclease activity of mainly double-stranded DNA, some interference with respiration and inhibition of RNA polymerase	[164,165,166]
Pyrithione	Copper complex and ionophore, facilitates copper influx	[167]
Tetracycline	Copper complex formation shows an antagonistic effect while complex formation with Cd(II) exhibits a synergistic effect	[168]
Fluoroquinolone	Copper complex formation but with only minor antimicrobial effect, facilitates the transfer of copper across hydrophobic membranes	[169,170]
Aminoglycoside	A weak copper complex formation that is not physiologically relevant, maybe a potential role in DNA damage and formation of reactive oxygen species	[171]

## Data Availability

Publicly available datasets were analyzed in this study. This data can be found here: https://www.rcsb.org/; https://www.uniprot.org/; https://www.ncbi.nlm.nih.gov/.

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
