# Peer review of "Advances in Understanding of the Copper Homeostasis in Pseudomonas aeruginosa"

_ijms, 2021, doi:10.3390/ijms22042050_

Round 1

Reviewer 1 Report

The manuscript deals with the P.aeruginosa Cu homeostasis and describes the multiple strategies that P aeruginosa uses for uptake, storage and export of Cu. A section of the manuscript is also dedicated to some structural insights into proteins involved in Copper homeostasis

The authors also claim: “we emphasize the use of heavy metals as antimicrobial agents also in combination with other antibiotics to overcome the emerging broad-spectrum resistance of P. aeruginosa.”

Some recent review has been published on the main topic of this review (Andrei et al., 2020), a point of strength of the paper is the attempt to compare the features of copper homeostasis in P.aeruginosa with those of E. coli. This could justify the work. However, some points of weakness are present: i) no attempt is made to distinguish the antimicrobial activity of copper in the two bacteria and the involvement of different players of export system; ii) the section 5 and 6 do not describe the mechanisms of action of the cited antimicrobial agents and their synergy with copper.

Minor points:

Line 29. “2.1” should be substituted by 1.1

Line 320. “the oxidation of Fe(II) to Fe(III) by reducing Cu(I) to Cu(II)”, this statement is not right

The manuscript cannot be recommended for publication in this version

Author Response

Reviewer (I)

The manuscript deals with the P.aeruginosa Cu homeostasis and describes the multiple strategies that P aeruginosa uses for uptake, storage and export of Cu. A section of the manuscript is also dedicated to some structural insights into proteins involved in Copper homeostasis

The authors also claim: “we emphasize the use of heavy metals as antimicrobial agents also in combination with other antibiotics to overcome the emerging broad-spectrum resistance of P. aeruginosa.”

Some recent review has been published on the main topic of this review (Andrei et al., 2020), a point of strength of the paper is the attempt to compare the features of copper homeostasis in P.aeruginosa with those of E. coli. This could justify the work. However, some points of weakness are present: i) no attempt is made to distinguish the antimicrobial activity of copper in the two bacteria and the involvement of different players of export system;

We thank the reviewer for the helpful comment. Therefore, we added reference 62 of Andrei et al., 2020. Moreover, we do anticipate that the antimicrobial activity of copper is in both bacteria identical. However, we emphasize differences of exporting pumps and key players involved in the copper homeostasis explicitly by adding Table 1). Also we added the following statement in addition to the table: Lines: 243ff.
“Major differences of proteins involved in copper homeostasis between the two species were found in MT, CYTO C, RND, and Sidero. E. coli is lacking a MT, also it does not express any cytoplasmic copper chaperones, but it contains a dedicated cusCBA system for copper export. Moreover, the types of siderophores involved in copper homeostasis is reduced compared to P. aeruginosa (Table 1).”

We further discuss in detail the proposed cusCBA copper export system found in E. coli and demonstrate that it is not present in P. aeruginosa. This was emphasized by phylogenetic analysis (Figure 2) and structural modelling of mexPQ-opmE (Figure 4). In addition, we emphasize a potential role of czcCBA in copper export which needs to be confirmed. Lines 343ff. and lines 444ff

  1. ii) the section 5 and 6 do not describe the mechanisms of action of the cited antimicrobial agents and their synergy with copper.

We thank the reviewer for the comment, thus we added the following paragraph to section 5, explaining the biocidal mechanism of copper: Lines 497ff.
“The antimicrobial, antifungal, and antiviral activity or in short biocidal activity of copper is based on two mechanisms (156). These two mechanisms are membrane damage, and oxidative stress that eventually results in destruction of genetic material (156–158). The ability of copper to damage both DNA and RNA through formation of reactive oxygen species also explains its antiviral activity (149,159).”

In addition, Table 3) describes the cooperative effect between copper and the drug. We termed this synergy between copper and drug “copperaction”. Further, we refer with at least one reference to the experimental paper that describes the mechanism in more detail. An extensive discussion of all mechanism of action would be out of scope. But we emphasized, the synergistic action of copper and Ditiocarb by proposing an alternative mechanism. There we propose that Ditiocarb reacts with the cysteine residues as suicide inhibitor. Thus, preventing copper binding of the two proteins csoR and ricR. This controversial perspective of the mechanism of action of Ditiocarb might provide an interesting avenue for future drug design. Moreover, we added a succinct description of the involved mechanisms of action for all listed combinations in addition to Table 3. For clarification, the following paragraph was added: Lines 524ff.

“In general, it was noted that all compounds listed in Table 3 form a complex with copper. Hence, the listed drugs in Table 3 are capable delivering copper into the pathogen cell as copper ionophore. Delivery of both the drug and copper will stress bacteria in two ways. First, it will increase the amount of reactive oxygen species within the cell. Second, the antibiotic will challenge the bacteria through the antibiotic specific mode of action. The copperaction of Ditiocarb was heavily debuted and assigned to the copper toxicity derived from macrophages (85). Here we disagree with this explanation and propose that Ditiocarb might interact with csoR or ricR and bind to the cysteine residues found at the active site. Thus, Ditiocarb might increase the efficacy of copper by delivering copper and act as suicide inhibitor by potentially linking the two cysteine residues at the metal binding site of csoR or ricR. 8-Hydroxyquinoline (8HQ) is a bidentate metal chelator capable of forming Cu(II) complex with different stoichiometry (e.g. Cu(II) : 8HQ; 1:1 and 1:2 complex) (175). The Cu(II) complex acts as ionophore and increases the cell associated labile copper ions, ultimately facilitating the destruction of Mycobacterium tuberculosis in primary macrophages. Also, it was shown that only the 1:1 stoichiometry shows antibacterial activity in Mycobacterium tuberculosis (165). Thiosemicarbazones are tetradentate chelators capable of complexing Cu(II) and act as ionophore similar to 8HQ but contain rotable bonds increasing the flexibility of the scaffold (164,176). Thiosemicarbazones themselves, are thought to impair respiratory enzymes such as dehydrogenases (166,177). In addition, it was demonstrated that these class of molecules also have activity independent of the redox cycle. Thus, it might be possible that Thiosemicarbazones also impair copper binding proteins via cysteine residues and their two sulfur groups as described above. Phenanthroline are bidental chelators known for their membrane permeability. In addition to the copper transport, phenanthrolines are capable of DNA and scission and inhibition of RNA polymerase resulting in their enhanced antimicrobial activity (169). Pyrithione is a bidental chelator similar to 8HQ but with a four-times higher potency than 8HQ (178). Its activity is solely based on the activity as copper ionophore without any biocidal activity itself (170). The classic antibiotics Tetracycline, Fluoroquinolone, and Aminoglycosides are bidental chelators showing copper ionophore activity in a 1:1 or 1:2 (metal : ligand) complex, in addition to their antimicrobial mechanism of action (85). In summary, the combination of the above mentioned antibiotics with copper displays a prototypical use of combinatorial drug treatment against human pathogens such as P. aeruginosa. A more detailed explanation of the synergistic mode of action can be found in the book of R. Poole et al. (85).”

Minor points:

Line 29. “2.1” should be substituted by 1.1

We thank the reviewer for the comment the numbering was adjusted.

Line 320. “the oxidation of Fe(II) to Fe(III) by reducing Cu(I) to Cu(II)”, this statement is not right

We thank the reviewer for the comment this misconception was corrected. Lines 338ff.

“Furthermore, homology search indicates that pcoA catalyzes the oxidation of Fe(II) to Fe(III), comparable to the oxidoreductase Fet3p found in Saccharomyces cerevisiae (93,101). Additionally, knock out studies demonstrated that pcoA is required for iron uptake in P. aeruginosa and is indeed a ferroxidase (133). But how these two proteins pcoA and pcoB interact with each other, including the detailed mechanisms of the copper redox reaction remains elusive.”

Reviewer 2 Report

The manuscript provides a nice overview of Pseudomonas aeruginosa and describes the important role that copper plays in Pseudomonas survival, antimicrobial resistance and homeostasis. Overall, the review was well written and easy to follow. The figures added to the manuscript and are detailed.

General comments:

Abstract is not accurately describing the content of the review?  The abstract spends focuses on background information about Pseudomonas aeruginosa. There is no mention of E. coli even though it is discussed in fair depth within the review. Only the last sentence of the abstract actually touches on the content of the review (Copper homeostasis). As such the abstract is unacceptable and needs complete revision to more accurately describe the contents of the review and match the title of the manuscript

lack of consistency between Copper and Cu, same for Zinc and Zn. Authors need to edit for consistency

Authors also switch between Cu(II) and Cu2+, edit for consistency

Specific comments:

Abstract: it is unclear what authors mean by "2.8 Mio"??

Line 52: "Such environment" should be "Such an environment"

Lines 83 and 84: P. aeruginosa should be in italics

Line 88: bacteria are plural, "bacteria is" should be "bacteria are"

Lines 91, 93, 96:  P. aeruginosa should be in italics

Lines 105-106: Please add a couple sentences and references about how QS has been explored as a potential target for therapeutics.  There has been much work done in this area and adding a few sentences will enhance the paragraph and importance of the QS system

Line 116: "The transported good"? This is confusing. Are the authors referring to LPS?

Line Lines 188-121: This section is confusing. Authors should consider revising.  A better description of how LPS enhances biofilm formation and increased antimicrobial resistance is needed.  Authors should also add a couple specific examples to help with this statement.

Line 193: Authors say C(II) but do they mean Cu(II)?

Line 198: Rhodobater capsulatus should be in italics

Line 311: Adjust column widths in table so that "reference" is in one line and not split

Author Response

Reviewer (II)

The manuscript provides a nice overview of Pseudomonas aeruginosa and describes the important role that copper plays in Pseudomonas survival, antimicrobial resistance and homeostasis. Overall, the review was well written and easy to follow. The figures added to the manuscript and are detailed.

General comments:

Abstract is not accurately describing the content of the review?  The abstract spends focuses on background information about Pseudomonas aeruginosa. There is no mention of E. coli even though it is discussed in fair depth within the review. Only the last sentence of the abstract actually touches on the content of the review (Copper homeostasis). As such the abstract is unacceptable and needs complete revision to more accurately describe the contents of the review and match the title of the manuscript

We thank the reviewer for the suggestion. The abstract was revised and adjusted according to the well observed discrepancy between abstract and content.

lack of consistency between Copper and Cu, same for Zinc and Zn. Authors need to edit for consistency

We thank the reviewer for the comment and changed Zinc to Zn throughout the manuscript. Regarding the copper nomenclature. Wherever the oxidation state of copper was crucial it was written as symbol (e.g. Cu(I) or Cu(II)). In case the oxidation state is not of importance “copper” was used instead of the symbols.

Authors also switch between Cu(II) and Cu2+, edit for consistency

We thank the reviewer for the comment and changed Cu2+ to Cu(II).

Specific comments:

Abstract: it is unclear what authors mean by "2.8 Mio"??

We thank the reviewer for the comment and corrected the sentence accordingly.
 “… more than 2.8 Mio million antibiotic resistant infections …”

Line 52: "Such environment" should be "Such an environment"

We thank the reviewer for the comment and corrected the sentence to “Such an environment is ….”.

Lines 83 and 84: P. aeruginosa should be in italics

We thank the reviewer for the comment and adjusted the font to italics.

Line 88: bacteria are plural, "bacteria is" should be "bacteria are"

We thank the reviewer for the comment and corrected the sentence to “The bacteria are multiplying….”.

Lines 91, 93, 96:  P. aeruginosa should be in italics

We thank the reviewer for the comment and adjusted the font to italics.

Lines 105-106: Please add a couple sentences and references about how QS has been explored as a potential target for therapeutics.  There has been much work done in this area and adding a few sentences will enhance the paragraph and importance of the QS system

We thank the reviewer for the suggestion and extended the paragraph with the following sentences. Lines 107ff.
“These targets are of tremendous interest since their crucial role in cell-cell communication and pathogenicity of P. aeruginosa (21,32,33). A screening of 12 active antibiotics revealed that indeed three of them (azithromycin, ceftazidime, and ciprofloxacin) decrease the expression of QS-regulated virulence factors (34). The reason for this decrease might be found in changes of the membrane permeability for certain QS factors (34). Moreover, a structure-based virtual screening approach against QS receptor lasR resulted in five active compounds capable of inhibiting QS regulated gene expression in P. aeruginosa (35). Using such compounds in combination with classic antibiotics are a promising strategy to overcome the notoriously resistant pathogen P. aeruginosa (36). Therefore, using signaling molecules of QS or a derivate thereof as Trojan horse to overcome multidrug-resistant P. aeruginosa provides a powerful strategy in the fight against pathogen bacteria.”

Line 116: "The transported good"? This is confusing. Are the authors referring to LPS?

We thank the reviewer for the suggestion and changed the sentence to: Line 124
These LPS are then transferred to ….”

Line Lines 188-121: This section is confusing. Authors should consider revising.  A better description of how LPS enhances biofilm formation and increased antimicrobial resistance is needed.  Authors should also add a couple specific examples to help with this statement.

We thank the reviewer for the suggestion and added the following lines to the existing paragraph. Line 126ff.
“In addition, exopolysaccharides such as Psl, Pel and extracellular DNA are abundant components of the biofilm in P. aeruginosa (45,46). These components adhere to each other and play an essential role in the highly complex biofilm formation, which is beneficial for growth and antibiotic tolerance (47–49). P. aeruginosa expresses soluble lectins lecA and lecB, both surface proteins capable of binding the exopolysaccharides (50). These binding molecules were both successfully targeted by glycopeptide dendrimers and monosaccharides disrupting the biofilm formation (50–54). Interestingly, it was shown that application of LPS from P. aeruginosa stabilize and increase biofilm formation of other Enterobacteriaceae (55). In conclusion, LPS contribute to a defense and communication system, while exopolysaccharides contribute to the biofilm formation. Hence, both providing substantially to the high virulence of P. aeruginosa.”

Line 193: Authors say C(II) but do they mean Cu(II)?

We thank the reviewer for the comment and corrected C(II) to Cu(II).

Line 198: Rhodobater capsulatus should be in italics

We thank the reviewer for the comment and adjusted the font to italics.

Line 311: Adjust column widths in table so that "reference" is in one line and not split

We thank the reviewer for the comment and adjusted the size of the Reference column.

Round 2

Reviewer 1 Report

The revised version of the manuscript can be recommended for publication